# Establishment of a two-dimensional PCR method for simultaneous detection of nine sexually transmitted disease pathogens: insights into coinfection rates and epidemiological trends in HPV screening

Shuang Yao,[1] Jun Zhang,[1] Lili Pan,[1] Yang Yu,[1] Guanghua Luo[1]

**ABSTRACT** This study developed a two-dimensional PCR (2D-PCR) method for simultaneous detection of nine sexually transmitted disease pathogens (STDPs) in cervical brush samples collected after human papillomavirus (HPV) screening, aiming to evaluate co-infection rates and epidemiological trends. The 2D-PCR assay was optimized under single-tube closed conditions, with sensitivity and accuracy validated. The prevalent nine STDPs were assessed in 2,193 females undergoing routine gynecological inspections and HPV screening. Statistical analysis revealed correlations between HPV genotypes and identified pathogens. Results showed detection limits (LODs) of $10^2$–$10^3$ copies/µL for STDPs, with high concordance to triplex real-time PCR (Kappa = 0.90). Overall, 48.6% of samples tested positive for ≥1 pathogen, with 36.02% positive for ≥1 STDP. The most prevalent STDPs were *Ureaplasma parvum*/*Ureaplasma urealyticum* (27.04%), *Mycoplasma hominis* (3.42%), and *Trichomonas vaginalis* (0.23%). HPV-positive individuals exhibited higher STDP infection rates (46.32%) than HPV-negative counterparts (32.86%). Significant associations were observed between HPV infection and *U. parvum*/*U. urealyticum* or *M. hominis*. HPV 52 was the predominant genotype in STDP-infected individuals (6.2%), with genotypes 52, 53, 6, 11, 42, 43, and 61 significantly more prevalent in STDP-positive groups. The 2D-PCR method proved sensitive and specific for multiplex STDPs detection, while epidemiological data highlighted significant STDP-HPV correlations.

**IMPORTANCE** This study introduces the first single-tube 2D-PCR method for efficient, high-throughput detection of nine STDPs, addressing a critical gap in co-infection diagnostics. The high prevalence of STDPs and their strong association with human papillomavirus (HPV) infection underscore the clinical relevance of co-pathogen screening, particularly in HPV-positive populations. The significant linkage between *U. parvum*/*U. urealyticum*, *M. hominis*, and HPV infection suggests potential synergistic mechanisms influencing HPV persistence or progression. Notably, the predominance of HPV 52 and increased prevalence of genotypes 53, 6, 11, 42, 43, and 61 in STDP-infected individuals highlight how pathogen co-infections may modulate HPV genotype distribution. These findings provide a robust tool for integrated STDPs/HPV screening and offer epidemiological insights to guide targeted prevention strategies, ultimately enhancing management of sexually transmitted infections and HPV-related cervical disease.

**KEYWORDS** 2D-PCR, sexually transmitted diseases, HPV, co-infection, epidemiology

**Peer Reviewers** Neena Abdul Abdul Hameed, Azeezia Medical College Hospital, Kollam, Kerala, India; Sabitha Baby, Karuna Medical College, Palakkad, Kerala, India

Address correspondence to Guanghua Luo, shineroar@163.com.

The authors declare no conflict of interest.

See the funding table on p. 11.

Sexually transmitted diseases (STDs) have a profound effect on reproductive and sexual health worldwide. New data on STDs from WHO show 374 million new cases per year, including 128 million cases of chlamydia, 82 million cases of gonorrhea, 156 million cases of trichomoniasis, and 7 million cases of syphilis (1). Sexually transmitted infections (STIs) continue to be a major public health burden in terms of mortality, morbidity, and quality of life, especially in developing countries.

The overall infection rate of high-risk human papillomavirus (HPV) in mainland Chinese women was 19% (2). With the global promotion and commercialization of the HPV vaccine, the importance of screening for HPV is increasingly recognized. Studies suggest that high-risk HPV is associated with the progression of cervical cancer, but it is not the only factor in the development of this disease (3). Chronic infections from other STIs, which lead to an inflammatory microenvironment, also contribute to the progression of cervical cancer and other reproductive system diseases (4). Therefore, this study has established an economical and simple detection method that repurposes "waste" cervical brush samples obtained after HPV screening. This method is capable of simultaneously testing nine types of STDPs from extracted DNA samples, including *Ureaplasma parvum/Ureaplasma urealyticum, Mycoplasma hominis, Trichomonas vaginalis, Mycoplasma genitalium, Neisseria gonorrhoeae, Chlamydia trachomatis, Herpes simplex virus type I* (HSV-1), and *Herpes simplex virus type II* (HSV-2). Patients or individuals undergoing health check-ups only require one sampling procedure to achieve concurrent detection of HPV and STDPs, reducing patient discomfort and significantly cutting down on economic costs. The detection method, based on the principles of 2D-PCR, enables single-tube simultaneous testing of nine STDPs. It is characterized by its simplicity, short reaction time, low cost, independence from additional product identification instruments, and avoidance of complex analysis procedures, while ensuring reliable and stable results.

Utilizing this method, this study conducted STDPs detection on population samples from the Changzhou region of Jiangsu Province, China, who underwent HPV screening between 2022 and 2023. The aim was to assess the co-infection rate of STDPs and HPV in this region and to analyze the epidemiological trends of STD in the population. These data provide a scientific basis for the development of effective prevention and treatment strategies for HPV and other STIs.

## RESULTS

### Establishment of 2D-PCR method for single-tube detection of nine STDPs

In this study, we developed and tested a 2D-PCR method targeting nine different etiological agents simultaneously to identify the most common STI-related pathogens, using hemoglobin subunit beta and hemoglobin subunit delta genes (*HBB&HBD*) as an internal control. The *HBB&HBD* positive control sample was derived from DNA extracted from human whole blood. Plasmids of eight positive reference strains, each with a concentration of $10^6$ copies/μL, were mixed with human whole blood DNA according to their respective detection channels. This mixture served as the amplification template to simulate multiple infections for constructing and optimizing the methodology. As shown in Fig. 1A, the FAM channel exhibits three distinct melting peaks at different temperatures, corresponding to *M. hominis* (46℃), *M. genitalium* (55.6℃), and *N. gonorrhoeae* (61.2℃), respectively. Fig. 1B shows the HEX channel with five distinct melting peaks at different temperatures, corresponding to *T. vaginalis* (44℃), HSV-1 (49.2℃), *HBB&HBD* (54.8℃), *C. trachomatis* (62℃), and HSV-2 (66.4℃). Fig. 1C shows the ROX channel with a melting peak at 61.2℃, corresponding to *U. parvum/U. urealyticum*. Fig. 1D shows the melting curve obtained using the 2D-PCR method to detect a cervical brush sample co-infected with *M. hominis* and *T. vaginalis*. Fig. 1E represents a cervical brush sample positive for *C. trachomatis*, and Fig. 1F depicts a cervical brush sample co-infected with *N. gonorrhoeae* and *U. parvum/U. urealyticum*.

The analytical sensitivity of the 2D-PCR method in this study was determined by testing serial dilutions of positive standards ranging from $10^5$ to $10^1$ copies/μL of

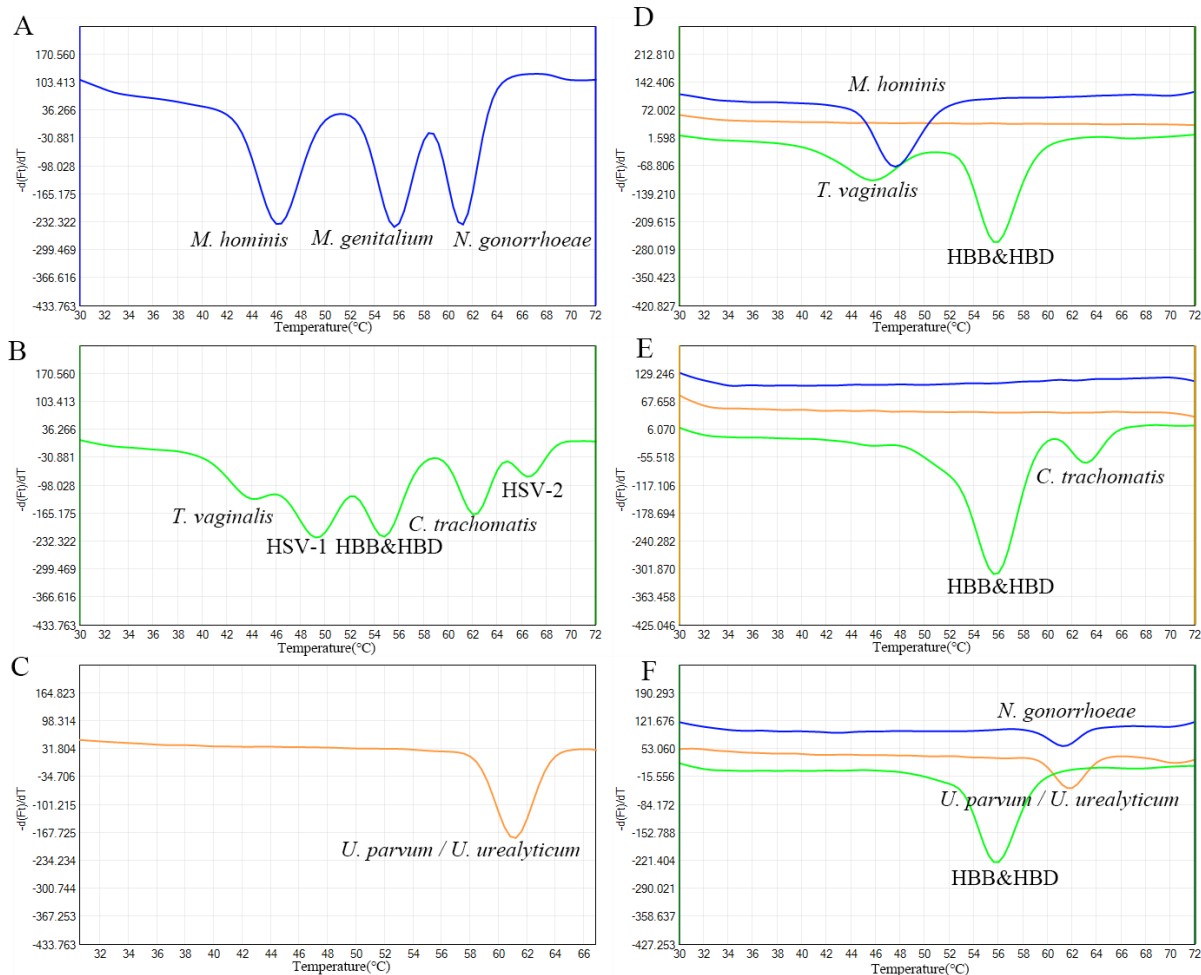

**FIG 1**  (A-C) Melting curves from mixed plasmids in FAM (A), HEX (B), and ROX (C) detection channels. (D-F) Clinical samples show co-infections with M. hominis & T. vaginalis (D), C. trachomatis (E), and N. gonorrhoeae & U. parvum/U. urealyticum (F).

plasmids carrying the target genes of each agent. As shown in Fig. 2, with the decreasing concentration of plasmids, the depth of each melting peak gradually becomes shallower. The melting peaks of all nine STDPs can still be clearly distinguished at concentrations of either $10^2$ or $10^3$ copies/µL. Among them, the LOD for *M. hominis, M. genitalium, N. gonorrhoeae,* and *U. parvum/U. urealyticum* is $10^2$ copies/µL, while the LOD for *T. vaginalis*, HSV-1, *C. trachomatis*, and HSV-2 is $10^3$ copies/µL.

## Analysis of consistency in identification results between 2D-PCR and triple real-time PCR

A total of 2,193 cervical brush samples were tested using both the 2D-PCR method and the triplex real-time fluorescence quantitative PCR method. Consistency analysis was performed on the results obtained from the two methods. As shown in Table 1, the overall detection consistency between the two methods is very high, with a kappa value of 0.90. Specifically, the kappa value for *U. parvum/U. urealyticum* is 0.91, for *M. hominis* is 0.98, for *T. vaginalis* is 0.91, for *M. genitalium* is 0.82, for *N. gonorrhoeae* is 1, for *C. trachomatis* is 1, and for HSV-2 is 1.

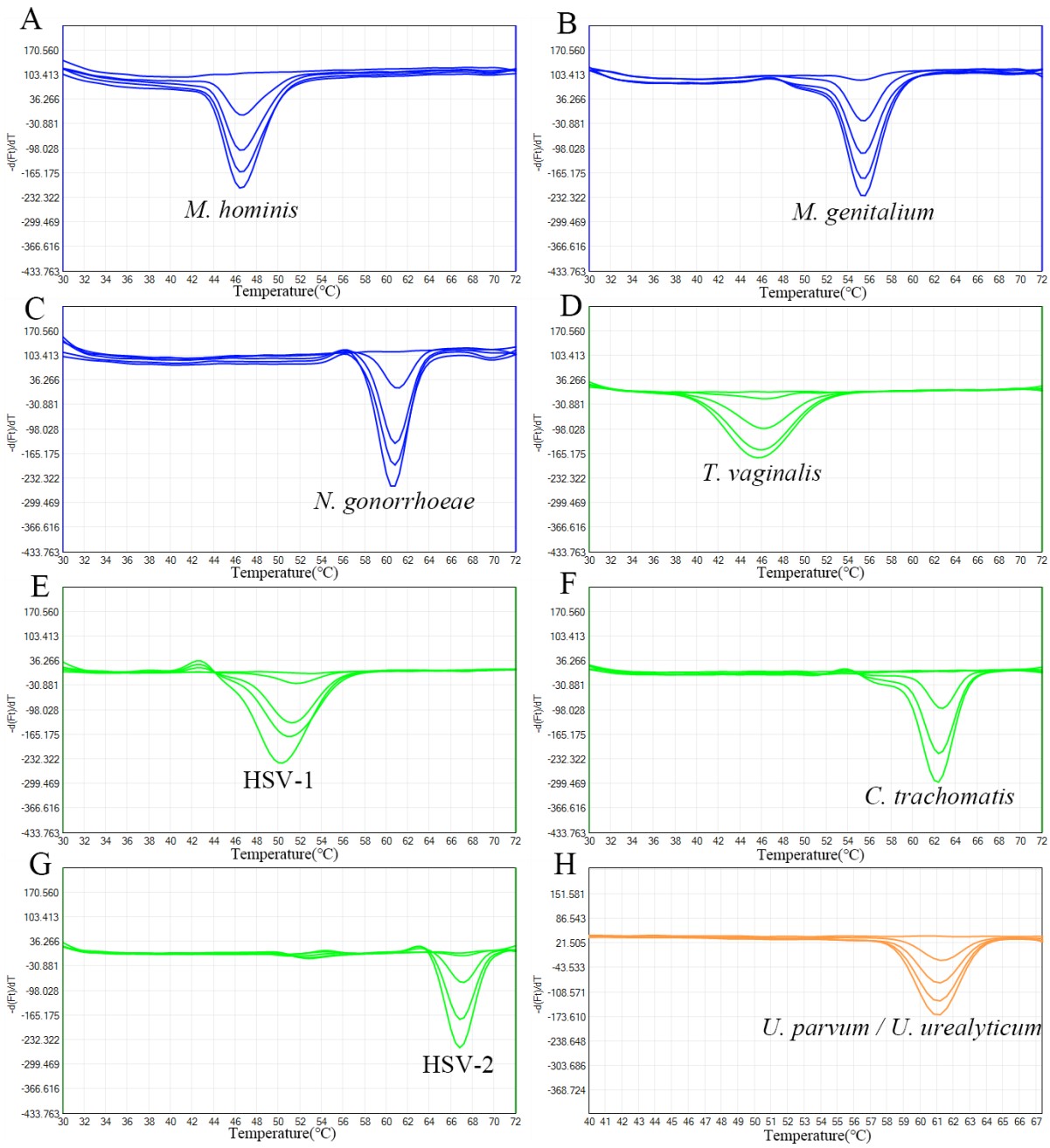

**FIG 2** Melting curves of pathogen-specific plasmids: M. hominis (A), M. genitalium (B), N. gonorrhoeae (C), T. vaginalis (D), HSV-1 (E), C. trachomatis (F), HSV-2 (G), U. parvum / U. urealyticum (H) at serial dilutions ($10^5$–$10^1$ copies/μL).

## Epidemiological investigation of co-infection of STDPs and HPV

A statistical analysis was conducted on the STDPs and HPV infection status in 2,193 cervical brush samples. The median age was 45 years (interquartile range [IQR], 35–53). A total of 48.6% (1,066/2,193) tested positive for one or more pathogens, while 51.4% (1,127/2,193) tested negative. One or more STDPs were detected in 36.02% (790/2,193) of patients, with a median age of 44 years (IQR 33.75–51). Specifically, the proportion of individuals with a single infection of *U. parvum/U. urealyticum* is 27.04% (593/2,193); *M. hominis* is 3.42% (75/2,193); *T. vaginalis* is 0.23% (5/2,193); and *M. genitalium*, *C. trachomatis*, or HSV-2 is 0.05% (1/2,193). STI co-infections represented 5.20% (114/2,193) of cases: *U. parvum/U. urealyticum* and *M. hominis* (n = 84), *U. parvum/U. urealyticum* and

**TABLE 1** Analysis of consistency in identification results between 2D-PCR and triple real-time PCR

| STDPs | 2D-PCR | Triple real-time PCR | | Kappa |
|---|---|---|---|---|
| | | Positive | Negative | |
| U. parvum/U. urealyticum | Positive | 696 | 4 | 0.91 |
| | Negative | 81 | 1,412 | |
| M. hominis | Positive | 165 | 7 | 0.98 |
| | Negative | 0 | 2,021 | |
| T. vaginalis | Positive | 20 | 4 | 0.91 |
| | Negative | 0 | 2,169 | |
| M. genitalium | Positive | 7 | 0 | 0.82 |
| | Negative | 3 | 2,183 | |
| N. gonorrhoeae | Positive | 2 | 0 | 1 |
| | Negative | 0 | 2191 | |
| C. trachomatis | Positive | 3 | 0 | 1 |
| | Negative | 0 | 2,190 | |
| HSV-2 | Positive | 2 | 0 | 1 |
| | Negative | 0 | 2,191 | |
| Total | Positive | 755 | 14 | 0.90 |
| | Negative | 85 | 1,339 | |

*T. vaginalis* (*n* = 8), and *M. hominis* and *T. vaginalis* (*n* = 7). The distribution of pathogenic microorganisms among populations infected with STDPs (n=790) is shown in Fig. 3A.

Fig. 3B illustrates the age distribution curves of the three most prevalent STDPs (*U. parvum/U. urealyticum*, *M. hominis*, and *T. vaginalis*) and HPV-positive individuals. The age-stratified analysis shows that the populations infected with *U. parvum/U. urealyticum* and *T. vaginalis* are primarily within the age bracket of 41 to 50 years. The population infected with *M. hominis* is evenly distributed across the age ranges of 31–40 and 51–60 years. The HPV-infected population is predominantly within the age range of 31 to 40 years.

Table 2 shows the correlation between STDPs and HPV infections. Among 2,193 patients, 516 were HPV positive (23.53%), while 1,677 were HPV negative (76.47%). Among the HPV-positive patients, 239 were also infected with STDPs, accounting for 46.32% of the total number of HPV-positive cases. In the HPV-negative population, the number of individuals carrying STDPs was 551, accounting for 32.86% of the HPV-negative group. Therefore, the proportion of HPV-positive patients carrying STDPs is higher than that in the HPV-negative population. Specifically, *U. parvum/U. urealyticum* and *M. hominis* are associated with HPV infection. In the HPV-positive group, 39.92% carried *U. parvum/U. urealyticum*, and 14.1% carried *M. hominis*. In contrast, infections with *T. vaginalis* and *M. genitalium* showed no significant correlation with HPV infection.

At the same time, we analyzed the correlation between multiple STDP infections and multiple HPV infections. Among the 239 patients simultaneously infected with STDPs and HPV, 14 (5.85%) samples were simultaneously infected with multiple HPVs and multiple STDPs, 145 (60.67%) samples were infected with a single HPV and a single STDP, 41 samples were infected with multiple HPVs while also being infected with one STDP, and 30 samples were infected with multiple STDPs while also being infected with one HPV type. Statistical analysis indicates that there is no significant correlation between multiple STDP infections and multiple HPV infections (Table S5).

Table 3 shows the correlation between different types of STDPs and high-risk (HR) and low-risk (LR) HPV infections. Among them, the infection of *U. parvum/U. urealyticum* was associated with high-risk or low-risk HPV infections. Among the 516 HPV-positive cases, there were 353 cases of HR-HPV infection, 110 cases of LR-HPV infection, and 53 cases with co-infection of both high-risk and low-risk HPV (HLR-HPV). Among the HR-HPV-infected individuals, 124 cases (35.13%) were also infected with *U. parvum/U. urealyticum*. In the LR-HPV-infected group, 55 cases (50%) were co-infected with *U. parvum/U. urealyticum*. Among patients with co-infection of HLR-HPV, 27 cases (50.94%)

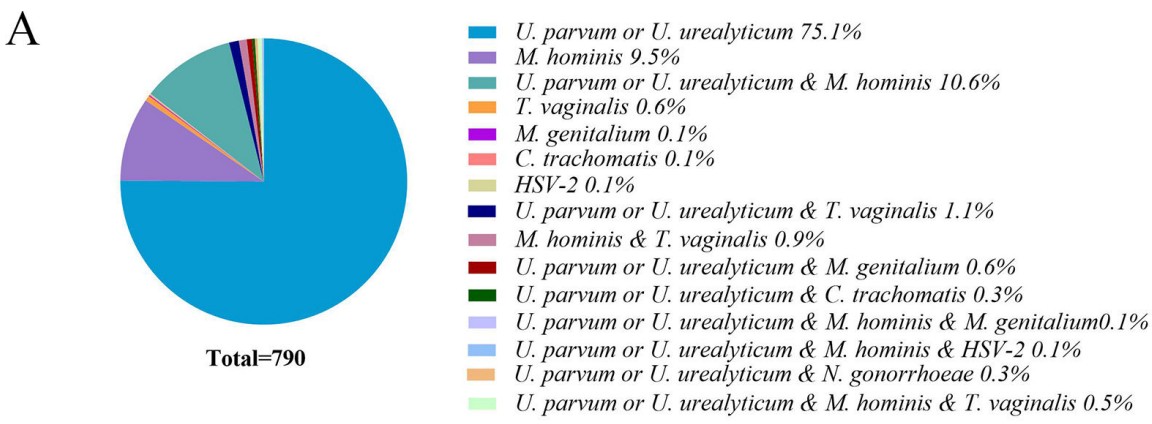

A

- U. parvum or U. urealyticum 75.1%
- M. hominis 9.5%
- U. parvum or U. urealyticum & M. hominis 10.6%
- T. vaginalis 0.6%
- M. genitalium 0.1%
- C. trachomatis 0.1%
- HSV-2 0.1%
- U. parvum or U. urealyticum & T. vaginalis 1.1%
- M. hominis & T. vaginalis 0.9%
- U. parvum or U. urealyticum & M. genitalium 0.6%
- U. parvum or U. urealyticum & C. trachomatis 0.3%
- U. parvum or U. urealyticum & M. hominis & M. genitalium 0.1%
- U. parvum or U. urealyticum & M. hominis & HSV-2 0.1%
- U. parvum or U. urealyticum & N. gonorrhoeae 0.3%
- U. parvum or U. urealyticum & M. hominis & T. vaginalis 0.5%

Total=790

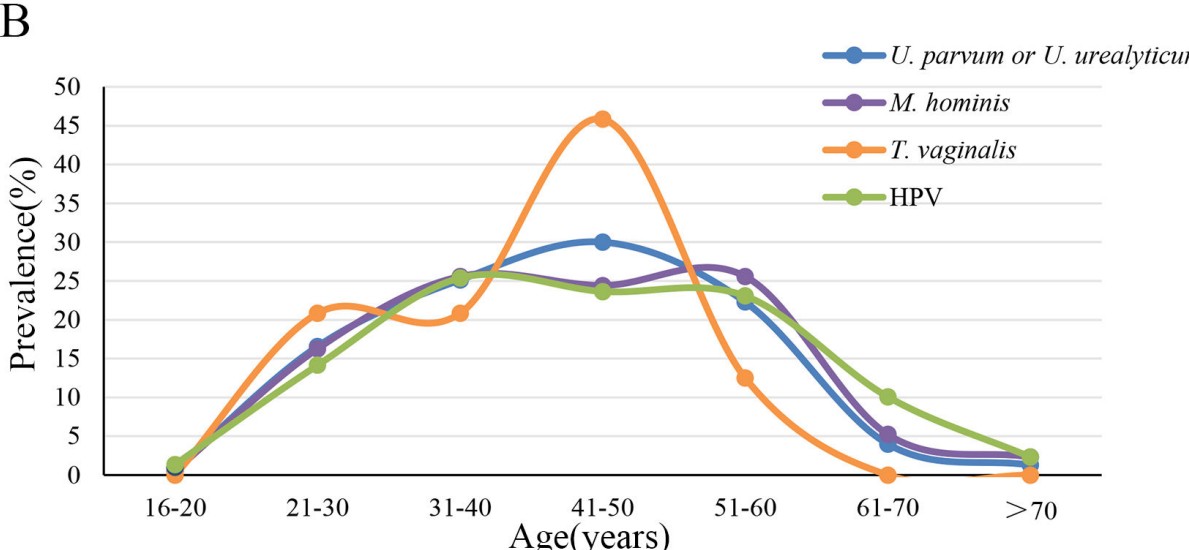

B

**FIG 3** (A) Pie chart showing distribution of pathogenic microorganisms in 790 STDPs-positive patients. (B) The age distribution curves of the three most prevalent STDPs (U. parvum/U. urealyticum, M. hominis, and T. vaginalis) and HPV-positive individuals.

were also infected with *U. parvum/U. urealyticum*. Therefore, the prevalence of *U. parvum/U. urealyticum* was higher in the LR-HPV group and among those with co-infection of HLR-HPV.

The prevalence of genotypes in individuals with and without concomitant STDP infections is shown in Table 4. HPV 52 was the most frequent HPV type in individuals with concomitant STDPs, accounting for 6.2% (49/790), followed by HPV 61 at 4.6% (36/790) and HPV 53 at 3.9% (31/790). On the other hand, in individuals without concomitant STDP infections, HPV 52 was also the most common genotype, accounting for 3.8% (53/1,403), followed by HPV 16 at 2.4% (33/1,403) and 61 at 2.2% (31/1,403). HPV 52, 53, 6, 11, 42, 43, and 61 were significantly more prevalent in individuals with concomitant STDP infections ($P < 0.05$).

## DISCUSSION

According to the 2021 policy released by the Chinese government, efforts will be made to progressively increase the cervical cancer screening coverage rate, aiming for over 50% among eligible women by the end of 2025 (5). Recent expert consensus and guidelines recommend HPV genotyping as the primary screening method for cervical

**TABLE 2** Correlation analysis of different STDPs and HPV co-infection

| STDPs | | HPV positive (*n* = 516) | HPV negative (*n* = 1677) | $\chi^2$ | *P*-value |
|---|---|---|---|---|---|
| *U. parvum/* | Positive | 206 (39.92%) | 494 (29.50%) | 19.89 | **<0.0001**[a] |
| *U. urealyticum* | Negative | 310 (60.08%) | 1,183 (70.50%) | | |
| *M. hominis* | Positive | 73 (14.10%) | 99 (5.90%) | 37.10 | **<0.0001**[a] |
| | Negative | 443 (85.90%) | 1,578 (94.10%) | | |
| *T. vaginalis* | Positive | 7 (1.40%) | 17 (1.00%) | 0.43 | 0.5127 |
| | Negative | 509 (98.60%) | 1,660 (99.00%) | | |
| *M. genitalium* | Positive | 2 (0.39%) | 5 (0.30%) | 0.10 | 0.7528 |
| | Negative | 514 (99.61%) | 1,672 (99.70%) | | |
| *N. gonorrhoeae* | Positive | 2 (0.39%) | 0 (0%) | –[b] | 0.0553 |
| | Negative | 514 (99.61%) | 1,677 (100%) | | |
| *C. trachomatis* | Positive | 1 (0.19%) | 2 (0.12%) | –[b] | 0.5530 |
| | Negative | 515 (99.81%) | 1,675 (99.88%) | | |
| HSV-2 | Positive | 1 (0.19%) | 1 (0.06%) | –[b] | 0.4153 |
| | Negative | 515 (99.81%) | 1,676 (99.94%) | | |
| STDPs | Positive | 239 (46.32%) | 551 (32.86%) | 31.03 | **<0.0001**[a] |
| | Negative | 277 (53.68%) | 1,126 (67.14%) | | |

[a]*P* value less than 0.05 were bold.
[b]The "(-)" indicates that Fisher's exact test was used here, and the chi-square value cannot be provided.

cancer (6). Cervical cancer development is multifactorial, infections with other STDPs can disrupt the cervical microenvironment, increasing the risk of HPV infection and potentially promoting tumorigenesis alongside HPV (7). Therefore, co-testing for other STDPs with HPV screening would enable clinicians to provide precise and effective personalized treatments.

Most STDP diagnostic test kits on the market utilize PCR methods (8). However, many of these kits either detect a limited number of STDPs or require multiple reaction tubes for multiplex detection, inadvertently increasing testing costs. Our research group has developed a 2D-PCR technology to simultaneously identify multiple target genes or SNPs in a closed tube using base-quenched probe technology and fluorescence melting temperature analysis. In this research, we developed a single-tube assay capable of

**TABLE 3** Correlation analysis between infections of different STDPs and infections of high-risk and low-risk HPV types

| STDPs | | HR-HPV (*n* = 353) | LR-HPV (*n* = 110) | HLR-HPV (*n* = 53) | $\chi^2$ | *P*-value |
|---|---|---|---|---|---|---|
| *U. parvum/* | Positive | 124 (35.13%) | 55 (50.00%) | 27 (50.94%) | 10.73 | **0.0047**[a] |
| *U. urealyticum* | Negative | 229 (64.87%) | 55 (50.00%) | 26 (49.06%) | | |
| *M. hominis* | Positive | 45 (12.75%) | 18 (16.36%) | 10 (18.87%) | 1.986 | 0.3704 |
| | Negative | 308 (87.25%) | 92 (83.64%) | 43 (81.13%) | | |
| *T. vaginalis* | Positive | 4 (1.13%) | 3 (2.73%) | 0 (0%) | 2.405 | 0.3004 |
| | Negative | 349 (98.87%) | 107 (97.27%) | 53 (100%) | | |
| *M. genitalium* | Positive | 1 (0.28%) | 0 (0%) | 1 (1.89%) | 3.613 | 0.1642 |
| | Negative | 352 (99.72%) | 110 (100%) | 52 (98.11%) | | |
| *N. gonorrhoeae* | Positive | 1 (0.28%) | 1 (0.91%) | 0 (0%) | 1.081 | 0.5826 |
| | Negative | 352 (99.72%) | 109 (99.09%) | 53 (100%) | | |
| *C. trachomatis* | Positive | 0 (0%) | 1 (0.91%) | 0 (0%) | 3.698 | 0.1574 |
| | Negative | 353 (100%) | 109 (99.09%) | 53 (100%) | | |
| HSV-2 | Positive | 0 (0%) | 1 (0.91%) | 0 (0%) | 3.698 | 0.1574 |
| | Negative | 353 (100%) | 109 (99.09%) | 53 (100%) | | |
| STDPs | Positive | 147 (41.64%) | 61 (55.45%) | 31 (58.49%) | 9.954 | **0.0069**[a] |
| | Negative | 206 (58.36%) | 49 (44.55%) | 22 (41.51%) | | |

[a]*P* value less than 0.05 were bold.

**TABLE 4** Prevalence of HPV types in patients tested positive and negative for STDPs[a]

| HPV types | STDPs Positive (n = 790) | STDPs Negative (n = 1403) | $\chi^2$ | P-value |
|---|---|---|---|---|
| HPV16 | 28 | 33 | 2.656 | 0.1031 |
| HPV18 | 8 | 12 | 0.139 | 0.7098 |
| HPV31 | 5 | 6 | 0.115 | 0.7351 |
| HPV33 | 3 | 8 | 0.085 | 0.7708 |
| HPV35 | 6 | 5 | 0.937 | 0.3330 |
| HPV39 | 11 | 18 | 0.0464 | 0.8295 |
| HPV45 | 2 | 4 | 0.0831 | 0.7731 |
| HPV51 | 8 | 18 | 0.315 | 0.5745 |
| HPV52 | 49 | 53 | 6.701 | **0.0096[a]** |
| HPV53 | 31 | 25 | 9.32 | **0.0023[a]** |
| HPV56 | 11 | 16 | 0.264 | 0.6074 |
| HPV58 | 25 | 31 | 1.852 | 0.1735 |
| HPV59 | 14 | 24 | 0.011 | 0.9156 |
| HPV66 | 8 | 9 | 0.905 | 0.3414 |
| HPV68 | 4 | 6 | 0.005 | 0.9461 |
| HPV82 | 1 | 4 | 0.079 | 0.7788 |
| HPV6 | 13 | 7 | 7.353 | **0.0067[a]** |
| HPV11 | 5 | 1 | 3.966 | **0.0464[a]** |
| HPV40 | 2 | 0 | –[b] | 0.1297 |
| HPV42 | 4 | 0 | 4.607 | **0.0318[a]** |
| HPV43 | 13 | 7 | 7.353 | **0.0067[a]** |
| HPV44 | 10 | 13 | 0.5605 | 0.4541 |
| HPV55 | 5 | 6 | 0.1145 | 0.7351 |
| HPV61 | 36 | 31 | 9.403 | **0.0022[a]** |
| HPV81 | 8 | 5 | 2.664 | 0.1026 |
| HPV83 | 2 | 3 | 0.079 | 0.7788 |

[a]P value less than 0.05 were bold.
[b]The "(-)" indicates that Fisher's exact test was used here, and the chi-square value cannot be provided.

detecting 9 STDPs simultaneously with the 2D-PCR method, making concurrent and cost-effective screening of HPV and STDPs feasible.

After developing, optimizing, and validating the 2D-PCR detection method, we investigated the prevalence of STDPs in 2,193 individuals who underwent HPV screening. The results were compared with triplex real-time fluorescent quantitative PCR for consistency, showing high overall consistency (kappa = 0.90). For *U. parvum/U. urealyticum*, the sensitivity of 2D-PCR was inferior to triplex real-time PCR, with 81 positive cases by qPCR not detected by 2D-PCR. To enhance sensitivity for *U. parvum/U. urealyticum*, we conducted various optimizations, including designing specific primers, matching them with different tags, and assigning *U. parvum/U. urealyticum* to a separate channel to prevent probe competition during amplification. The concordance for detecting *M. hominis* and *T. vaginalis* was 0.98 and 0.91, respectively, with 2D-PCR showing superior sensitivity compared with qPCR. The kappa value for *M. genitalium* was 0.82, potentially due to the limited number of positive cases.

In our study, *U. parvum/U. urealyticum* were the predominant STDPs, representing 31.92% of cases, with *M. hominis* next at 7.84% (including multiple infections). A study from Shanghai (2016–2021) found a similar prevalence of *U. urealyticum* (9). *U. parvum* and *U. urealyticum* are associated with various clinical manifestations, notably adverse pregnancy outcomes like chorioamnionitis and preterm premature rupture of membranes leading to preterm birth (10). Evidence suggests a causal role for *U. parvum/U. urealyticum* in nongonococcal urethritis and male infertility (11). In Xi'an, China, the prevalence of *M. hominis* was 6.48% (12), similar to our results. *M. hominis*, as an endosymbiont of *T. vaginalis*, typically co-infects (13). In our study, of 24 *T.*

*vaginalis*-positive cases, 11 were also positive for *M. hominis*. *M. hominis* is linked to various diseases, including pelvic inflammatory disease, cervicitis, and pyelonephritis (14). Genital herpes, a chronic sexually transmitted infection caused by HSV-1 or HSV-2, is characterized by recurrent genital ulcers. While 2D-PCR is highly sensitive and specific, false-negative results can occur. For instance, swabs taken without genital ulcers may lack sensitivity due to intermittent genital HSV shedding (15). In our study, HSV-1 was not detected, while HSV-2 was found in only two cases. Another reason may be that patients with genital herpes symptoms often seek care at STD clinics rather than gynecological clinics.

Our study found higher infection rates of *U. parvum*/*U. urealyticum* and *M. hominis* in the HPV-positive population, consistent with previous findings (16, 17). Research indicates that these infections may increase HPV risk by affecting immune response balance or due to lifestyle factors, leading to higher HPV and STDP rates (18). *U. parvum* and *U. urealyticum* are also linked to HPV persistence and early cervical cytological changes (19). Therefore, treating these infections during HPV treatment may help prevent early cervical cancer progression. Our data showed no correlation between multiple STDP coinfections and multiple HPV genotypes, but *U. parvum*/*U. urealyticum* infection rates were higher in low-risk HPV populations, a novel finding requiring further investigation. We analyzed specific HPV genotypes and STDP infections, finding associations with HPV types 52, 53, 6, 11, 42, 43, and 61. Types 52 and 53 are HR-HPV, while the others are LR-HPV, which are less associated with cervical cancer than types 16 or 18, raising questions about whether STDP co-infections accelerate cervical lesion progression.

In fact, we also developed a 2D-PCR method for the identification and genotyping of 16 HR-HPV types and related tumor suppressor genes p53 and RB1 for cervical cancer (20). Therefore, using 2D-PCR as a cost-effective screening method that can simultaneously detect HPV and other STDPs would have significant clinical and economic value. This approach enables early detection, prevention, and management of these infections, leading to reduced disease burden and healthcare costs.

## MATERIALS AND METHODS

### Study population and clinical specimens

This study included 2,193 women who underwent routine gynecological inspections from November 2022 to March 2023 at the Third Affiliated Hospital of Soochow University. Inclusion criteria: women of reproductive age over 18 years old; having sexual experience; having regular menstruation; not using any medications within 1 week; and no vaginal douching, cervical treatment, or sexual intercourse within 72 h. Exclusion criteria: women who are pregnant or lactating and women with chronic diseases requiring long-term medication. Exfoliated cervical cells were obtained from the ecto- and endo-cervix portions of the uterus using a cytobrush. With approval from the Institutional Ethics Committee (approval number: [2022 (ke) No. 046]), residual cervical swab samples that could not be identified individually were used.

### HPV DNA detection and genotyping

HPV testing was conducted using the Tellgenplex HPV 27 genotyping assay (Tellgen Corporation, Shanghai, China) on the Luminex 200 platform (Luminex Corporation, Austin, TX). The assay is a flow cytometry fluorescence hybridization method that detects 17 high-risk HPV types (HPV16, 18, 26, 31, 33, 35, 39, 45, 51, 52, 53, 56, 58, 59, 66, 68, and 82) and 10 low-risk HPV types (HPV 6, 11, 40, 42, 43, 44, 55, 61, 81, and 83).

### STDP detection

The residual DNA from HPV genotyping was utilized for the detection of STDPs.

### Preparation of positive control plasmids

Plasmids containing the highly conserved sequence regions of the nine STDPs were synthesized by Sangon Biotech Co., Ltd. (Shanghai, China). The pUC57 vector containing the target fragments was cloned and amplified in *E. coli* JM109 cells, followed by extraction and purification.

### 2D-PCR primers and probes

Primers for the nine STDPs were designed using the software Primer Premier 5.0 (Premier Biosoft Intl., California, USA) and tested with NCBI BLAST to ensure specificity. According to the principles of 2D-PCR, a tag homologous to the probe was linked to the 5′ end of the forward primer, with several mismatched bases between the probe sequence and the complementary sequence of the tag. Only one probe is required for each detection channel. Three probes required for the detection of the nine STDPs were labeled with carboxyfluorescein (FAM), hexachloro-fuorescein (HEX), and Alexa 568, respectively. Both primers and probes were synthesized by Sangon Biotech Co., Ltd. (Shanghai, China). The primers and probes are listed in Table S1.

### 2D-PCR reaction

The formulation of the 2D-PCR reaction system is shown in Table S2. PCR amplifications and melting curve analyses were performed using a SLAN-96S real-time PCR machine (Hongshi Tech, Shanghai, China). Cycling conditions included preincubation at 95°C for 10 min, followed by amplification for a total of 40 cycles under the following conditions: denaturation at 95°C for 5 s and annealing at 60°C for 15 s. The fluorescence acquisition began with heating at 95°C for 30 s and then at 30°C for 4 min; the temperature was gradually increased from 30 to 72°C with a ramp rate of 0.06°C/s, during which the fluorescence signal was acquired continuously. Fluorescence intensity was measured using three detection channels: FAM, HEX, and ROX. Plasmids containing nine STDPs with a concentration of $10^6$ copies/μL and human whole blood DNA were thoroughly mixed according to their detection channels to simulate multiple infections as the amplification template for methodological optimization. The assay targets *HBB* and *HBD* as an internal control to monitor DNA purification efficiency, PCR inhibition, and cell adequacy.

### Triple RT-PCR primers and probes

Primers and probes for nine STDPs were designed using Primer Premier 5.0 (Premier Biosoft) and validated for specificity using NCBI BLAST. In each PCR reaction, three probes were labeled with FAM, ROX, and VIC fluorescent dyes, respectively. The primers and probes are listed in Table S3. The formulation of the reaction system is shown in Table S4. Amplification began with an initial denaturation step at 95°C for 10 min, followed by 40 cycles of denaturation at 95°C for 10 s and annealing/extension at 60°C for 15 s. Fluorescence acquisition started with a 4 min incubation at 30°C, followed by a gradual temperature increase from 30°C to 80°C at a ramp rate of 0.1°C/s, during which the fluorescence signal was continuously monitored. The final step involved cooling at 40°C for 30 s.

### Sensitivity experiments of the 2D-PCR detection system

To evaluate the sensitivity of the 2D-PCR detection system constructed in this study, the STD positive control plasmids were diluted with TE buffer to concentrations of $10^5$, $10^4$, $10^3$, $10^2$, and $10^1$ copies/μL. Then, 5 μL of the positive control plasmids was subjected to the 2D-PCR assay to analyze the lowest template concentration detectable by this method.

### Statistical analysis

Categorical variables were represented as proportions, while median and interquartile range (IQR) values were calculated for continuous variables. The consistency of the

two detection methods was assessed using the SPSSAU online analysis software by calculating the kappa value. The chi-squared test ($\chi^2$) was used to compare categorical variables across groups. The concordance rates were analyzed using the kappa test, with kappa values of 0.2, 0.2–0.4, 0.4–0.6, and >0.6 considered as poor, fair, moderate, and good agreement, respectively.

## ACKNOWLEDGMENTS

This research was supported by Natural Science Foundation of Jiangsu Province (grant number BK20211063 to L.G.H.), Changzhou Sci & Tech Program (grant number CJ20210113 to Y.S.), and Leading Talent of Changzhou "The 14th Five-Year Plan" High-Level Health Talents Training Project (grant no. 2022260 to L.G.H.) of China.

G.L.: Conceptualization, methodology, writing – reviewing and editing the manuscript. S.Y.: Writing – original draft, performed the experiments, investigation and analyzed the data. J.Z.: Validation, analyzed the data. Y.Y.: Probe library establishment. L.P.: Sample collection and analyzed the data. All the authors read and approved the final manuscript.

The authors: No reported conflicts of interest. All authors have submitted the ICMJE Form for Disclosure of Potential Conflicts of Interest. Conflicts that the editors consider relevant to the ontent of the manuscript have been disclosed.

## AUTHOR AFFILIATION

[1]Clinical Medical Research Center, The Third Affiliated Hospital of Soochow University, Changzhou, China

## AUTHOR ORCIDs

Shuang Yao ⓘ http://orcid.org/0000-0003-3686-5404
Yang Yu ⓘ http://orcid.org/0000-0002-9258-786X
Guanghua Luo ⓘ http://orcid.org/0000-0001-8339-2828

## FUNDING

| Funder | Grant(s) | Author(s) |
| --- | --- | --- |
| Natural Science Foundation of Jiangsu Province | BK20211063 | Guanghua Luo |
| Changzhou Sci & Tech Program | CJ20210113 | Shuang Yao |
| Leading Talent of Changzhou "The 14th Five-Year Plan" High-Level Health Talents Training Project | 2022260 | Guanghua Luo |

## AUTHOR CONTRIBUTIONS

Shuang Yao, Conceptualization, Data curation, Formal analysis, Funding acquisition, Investigation, Methodology, Project administration, Resources, Software, Supervision, Validation, Visualization, Writing – original draft, Writing – review and editing | Jun Zhang, Formal analysis, Methodology, Validation | Lili Pan, Data curation, Formal analysis, Methodology | Yang Yu, Data curation, Methodology | Guanghua Luo, Funding acquisition, Project administration, Supervision, Validation, Writing – review and editing

## DATA AVAILABILITY

Upon submission, authors agree to make any materials, data, and associated protocols available upon request.

## ADDITIONAL FILES

The following material is available online.

## Supplemental Material

**Supplemental material (Spectrum00237-25-s0001.pdf).** Tables S1 to S5.

## Open Peer Review

**PEER REVIEW HISTORY (review-history.pdf).** An accounting of the reviewer comments and feedback.

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
