## [Reviewer comments · Microbiology Spectrum]

Microbiology Spectrum

Establishment of a Two-Dimensional PCR Method for Simultaneous Detection of Nine Sexually Transmitted Disease Pathogens: Insights into Coinfection Rates and Epidemiological Trends in HPV Screening

Shuang Yao, Jun Zhang, Lili Pan, Yang Yu, and Ghuanghua Luo

Corresponding Author(s): Shuang Yao, The Third Affiliated Hospital of Soochow University

Review Timeline:

Submission Date:	January 23, 2025
Editorial Decision:	March 11, 2025
Revision Received:	March 12, 2025
Accepted:	March 25, 2025

Editor: Day-Yu Chao

Reviewer(s): Disclosure of reviewer identity is with reference to reviewer comments included in decision letter(s). The following individuals involved in review of your submission have agreed to reveal their identity: Neena Abdul Abdul Hameed (Reviewer #1); sabitha baby (Reviewer #2)

Transaction Report:

DOI: <https://doi.org/10.1128/spectrum.00237-25>

Re: Spectrum00237-25 (Establishment of a Two-Dimensional PCR Method for Simultaneous Detection of Nine Sexually Transmitted Disease Pathogens: Insights into Coinfection Rates and Epidemiological Trends in HPV Screening)

Dear Dr. Shuang Yao:

Thank you for the privilege of reviewing your work. Below you will find my comments, instructions from the Spectrum editorial office, and the reviewer comments.

Revision Guidelines

Sincerely,
Day-Yu Chao
Editor
Microbiology Spectrum

Reviewer #1 (Comments for the Author):

Good study, but needs revision regarding the 9 different pathogens

Reviewer #2 (Comments for the Author):

This study highlights the role molecular diagnostics can play in diagnosing , difficult to culture organisms

**Establishment of a Two-Dimensional PCR Method for Simultaneous Detection of Nine Sexually**
**Transmitted Disease Pathogens: Insights into Coinfection Rates and Epidemiological Trends in**
**HPV Screening**

Shuang Yao¹, Jun Zhang¹, Lili Pan¹, Yang Yu¹, Guanghua Luo¹

¹ Clinical Medical Research Center, The Third Affiliated Hospital of Soochow University,
Changzhou, China

Correspondence to: Dr. Guanghua Luo, Clinical Medical Research Center, The Third Affiliated
Hospital of Soochow University, No. 185 Juqian Street, Changzhou 213003, P.R. China. E-mail:
shineroar@163.com

**Word count: 3,544**

**Abstract**

**Objective** This study aimed to establish a two-dimensional PCR (2D-PCR) method for the
simultaneous detection of nine sexually transmitted disease pathogens (STDPs) in cervical brush
samples collected after HPV screening, to evaluate co-infection rates and epidemiological trends.

**Methods** A 2D-PCR methodology was developed under single-tube closed conditions to identify
nine STDPs, with sensitivity and accuracy evaluated. The prevalence of nine STDPs was assessed in
2,193 females undergoing routine gynecological inspections and HPV screening. Statistical analysis
revealed correlations between HPV genotypes and identified pathogens.

**Results** The 2D-PCR method demonstrated limits of detection (LOD) between 10^2 and 10^3 copies/ μ L
for various STDPs and showed high consistency with triplex real-time PCR ($\text{Kappa}=0.94$). Among
the samples, 48.6% tested positive for one or more pathogens, and 36.02% for one or more STDPs.
The most prevalent STDPs were *Ureaplasma parvum* (*U. parvum*) or *Ureaplasma urealyticum* (*U.*
*urealyticum*) (27.04%), followed by *Mycoplasma genitalium* (*M. genitalium*) (3.42%) and
*Trichomonas vaginalis* (*T. vaginalis*) (0.23%). The prevalence of HPV-positive patients with STDPs
(46.32%) was higher than that of HPV-negative patients (32.86%). *U. parvum* or *U. urealyticum* and
*M. hominis* were significantly associated with HPV infection. HPV 52 was the most common
genotype in those with STDP infections (6.2%), and genotypes 52, 53, 6, 11, 42, 43, and 61 were
significantly more prevalent in STDP-infected individuals compared to those without.

**Conclusions** The 2D-PCR method provides a sensitive and specific tool for simultaneous detection
of multiple STDPs in a single tube. Epidemiological findings show significant associations between
STDPs and HPV infection.

**Keywords** 2D-PCR, sexually transmitted diseases, HPV, co-infection, epidemiology

**1. Introduction**

Sexually transmitted diseases (STDs) have a profound effect on reproductive and sexual health
worldwide. New data on STDs from WHO show 374 million new cases per year, including 128
million cases of chlamydia; 82 million cases of gonorrhea; 156 million cases of trichomoniasis; and
7 million cases of syphilis[1]. Sexually transmitted infections (STIs) continue to be a major public
health burden in terms of mortality, morbidity, and quality of life, especially in developing countries.

The overall infection rate of high-risk HPVs in mainland Chinese women was 19%[2]. With the
global promotion and commercialization of the HPV vaccine, the importance of screening for HPV is
increasingly recognized. Studies suggest that high-risk HPV is associated with the progression of
cervical cancer, but it is not the only factor in the development of this disease[3]. Chronic infections
from other STIs, which lead to an inflammatory microenvironment, also contribute to the
progression of cervical cancer and other reproductive system diseases[4]. Therefore, this study has
established an economical and simple detection method that repurposes cervical brush "waste" after
HPV screening, allowing extracted DNA samples to be simultaneously tested for 9 STDs.
[revised manuscript text omitted]

- 10.3389/fpubh.2023.1228048. PubMed PMID: 38089034; PubMed Central PMCID: PMCPMC10711282.
- 10. Noda-Nicolau NM, Tantengco OAG, Poletini J, Silva MC, Bento GFC, Cursino GC, et al. Genital Mycoplasmas
and Biomarkers of Inflammation and Their Association With Spontaneous Preterm Birth and Preterm Prelabor Rupture of
Membranes: A Systematic Review and Meta-Analysis. *Front Microbiol.* 2022;13:859732. Epub 20220330. doi:
10.3389/fmicb.2022.859732. PubMed PMID: 35432251; PubMed Central PMCID: PMCPMC9006060.
- 11. Beeton ML, Payne MS, Jones L. The Role of *Ureaplasma* spp. in the Development of Nongonococcal Urethritis and
Infertility among Men. *Clin Microbiol Rev.* 2019;32(4). Epub 20190703. doi: 10.1128/cmr.00137-18. PubMed PMID:
31270127; PubMed Central PMCID: PMCPMC6750135.
- 12. Zeng XY, Xin N, Tong XN, Wang JY, Liu ZW. Prevalence and antibiotic susceptibility of *Ureaplasma urealyticum*
and *Mycoplasma hominis* in Xi'an, China. *Eur J Clin Microbiol Infect Dis.* 2016;35(12):1941-7. Epub 20160816. doi:
10.1007/s10096-016-2745-2. PubMed PMID: 27530531.
- 13. Margarita V, Congiargiu A, Diaz N, Fiori PL, Rappelli P. *Mycoplasma hominis* and *Candidatus Mycoplasma girerdii*
in *Trichomonas vaginalis*: Peaceful Cohabitants or Contentious Roommates? *Pathogens.* 2023;12(9). Epub 20230825. doi:
10.3390/pathogens12091083. PubMed PMID: 37764891; PubMed Central PMCID: PMCPMC10535475.
- 14. Yagur Y, Weitzner O, Barchilon Tiosano L, Paitan Y, Katzir M, Schonman R, et al. Characteristics of pelvic
inflammatory disease caused by sexually transmitted disease - An epidemiologic study. *J Gynecol Obstet Hum Reprod.*
2021;50(9):102176. Epub 20210601. doi: 10.1016/j.jogoh.2021.102176. PubMed PMID: 34087450.
- 15. Johnston C. Diagnosis and Management of Genital Herpes: Key Questions and Review of the Evidence for the 2021
Centers for Disease Control and Prevention Sexually Transmitted Infections Treatment Guidelines. *Clin Infect Dis.*
2022;74(Suppl_2):S134-s43. doi: 10.1093/cid/ciab1056. PubMed PMID: 35416970.
- 16. Ye H, Song T, Zeng X, Li L, Hou M, Xi M. Association between genital mycoplasmas infection and human
papillomavirus infection, abnormal cervical cytopathology, and cervical cancer: a systematic review and meta-analysis.

Arch Gynecol Obstet. 2018;297(6):1377-87. Epub 20180308. doi: 10.1007/s00404-018-4733-5. PubMed PMID:
29520664.

17. Klein C, Samwel K, Kahesa C, Mwaiselage J, West JT, Wood C, et al. Mycoplasma Co-Infection Is Associated with
Cervical Cancer Risk. *Cancers (Basel)*. 2020;12(5). Epub 20200428. doi: 10.3390/cancers12051093. PubMed PMID:
32353967; PubMed Central PMCID: PMC7281224.

18. Liang Y, Chen M, Qin L, Wan B, Wang H. A meta-analysis of the relationship between vaginal microecology,
human papillomavirus infection and cervical intraepithelial neoplasia. *Infect Agent Cancer*. 2019;14:29. Epub 20191026.
doi: 10.1186/s13027-019-0243-8. PubMed PMID: 31673281; PubMed Central PMCID: PMC6815368.

19. Xie L, Li Q, Dong X, Kong Q, Duan Y, Chen X, et al. Investigation of the association between ten pathogens
causing sexually transmitted diseases and high-risk human papilloma virus infection in Shanghai. *Mol Clin Oncol*.
2021;15(1):132. Epub 20210509. doi: 10.3892/mco.2021.2294. PubMed PMID: 34055347; PubMed Central PMCID:
PMC8138855.

20. Zhang J, Yao S, Yu Y, Yu M, Luo G. Development of a typing detection method for high-risk human papillomavirus
and related tumor suppressor genes p53 and RB1 based on two-dimensional PCR technology. *Chinese Journal of*
*Laboratory Medicine*. 2024;(04):391-400.

**Figure Legends**

**Figure 1. The 2D-PCR melting curves for the simultaneous detection of nine STDs in a single**
**tube.**

**Figure 2. Melting curves of different dilution plasmids for the detection of 9 STDs using the**
**2D-PCR method.**

**Figure 3. The distribution of pathogenic microorganisms among populations infected with STD**
**and their age stratification.**

**Table 1. Analysis of consistency in identification results between 2D-PCR and triple real-time**
 **PCR.**

STDPS	2D-PCR	Triple real-time PCR		Kappa
		Positive	Negative	
U. parvum / U. urealyticum	Positive	696	4	0.91
	Negative	81	1412	
M. hominis	Positive	164	5	0.98
	Negative	0	2024	
T. vaginalis	Positive	20	4	0.91
	Negative	0	2169	
M. genitalium	Positive	7	0	0.82
	Negative	3	2183	
N. gonorrhoeae	Positive	2	0	1
	Negative	0	2191	
C. trachomatis	Positive	3	0	1
	Negative	0	2190	
HSV-2	Positive	2	0	1
	Negative	0	2191	
Total	Positive	755	14	0.90
	Negative	85	1339	

**Table 2. Correlation analysis of different STDs and HPV co-infection.**

STDs		HPV Positive (n=516)	HPV Negative (n=1677)	χ^2	P -value
U. parvum/ U. urealyticum	Positive	206 (39.92%)	494 (29.50%)	19.89	P < 0.0001
	Negative	310 (60.08%)	1183 (70.50%)		
M. hominis	Positive	73 (14.10%)	99 (5.90%)	37.10	P < 0.0001
	Negative	443 (85.90%)	1578 (94.10%)		
T. vaginalis	Positive	7 (1.40%)	17 (1.00%)	0.43	P =0.5127
	Negative	509 (98.60%)	1660 (99.00%)		
M. genitalium	Positive	2 (0.39%)	5 (0.30%)	0.10	P =0.7528
	Negative	514 (99.61%)	1672 (99.70%)		
N. gonorrhoeae	Positive	2 (0.39%)	0 (0%)	-	P =0.0553
	Negative	514 (99.61%)	1677 (100%)		
C. trachomatis	Positive	1 (0.19%)	2 (0.12%)	-	P =0.5530
	Negative	515 (99.81%)	1675 (99.88%)		
HSV-2	Positive	1 (0.19%)	1 (0.06%)	-	P =0.4153
	Negative	515 (99.81%)	1676 (99.94%)		
STDs	Positive	239 (46.32%)	551 (32.86%)	31.03	P < 0.0001
	Negative	277 (53.68%)	1126 (67.14%)		

*P* value less than 0.05 were bold.

**Table 3. Correlation analysis between infections of different STDPs and infections of high-risk**
 **and low-risk HPV types.**

STDPs		HR-HPV (n=353)	LR-HPV (n=110)	HLR-HPV (n=53)	χ^2	P-value
U. parvum/	Positive	124 (35.13%)	55 (50.00%)	27 (50.94%)	10.73	P=0.0047
U. urealyticum	Negative	229 (64.87%)	55 (50.00%)	26 (49.06%)		
M. hominis	Positive	45 (12.75%)	18 (16.36%)	10 (18.87%)	1.986	P=0.3704
	Negative	308 (87.25%)	92 (83.64%)	43 (81.13%)		
T. vaginalis	Positive	4 (1.13%)	3 (2.73%)	0 (0%)	2.405	P=0.3004
	Negative	349 (98.87%)	107 (97.27%)	53 (100%)		
M. genitalium	Positive	1 (0.28%)	0 (0%)	1 (1.89%)	3.613	P=0.1642
	Negative	352 (99.72%)	110 (100%)	52 (98.11%)		
N. gonorrhoeae	Positive	1 (0.28%)	1 (0.91%)	0 (0%)	1.081	P=0.5826
	Negative	352 (99.72%)	109 (99.09%)	53 (100%)		
C. trachomatis	Positive	0 (0%)	1 (0.91%)	0 (0%)	3.698	P=0.1574
	Negative	353 (100%)	109 (99.09%)	53 (100%)		
HSV-2	Positive	0 (0%)	1 (0.91%)	0 (0%)	3.698	P=0.1574
	Negative	353 (100%)	109 (99.09%)	53 (100%)		
STDPs	Positive	147 (41.64%)	61 (55.45%)	31 (58.49%)	9.954	P=0.0069
	Negative	206 (58.36%)	49 (44.55%)	22 (41.51%)		

P value less than 0.05 were bold.

**Table 4. Prevalence of HPV types in patients tested positive and negative for STDPs.**

HPV types	STDPs Positive (n=790)	STDPs Negative (n=1403)	χ^2	P-value
HPV16	28	33	2.656	0.1031
HPV18	8	12	0.139	0.7098
HPV31	5	6	0.115	0.7351
HPV33	3	8	0.085	0.7708
HPV35	6	5	0.937	0.3330
HPV39	11	18	0.0464	0.8295
HPV45	2	4	0.0831	0.7731
HPV51	8	18	0.315	0.5745
HPV52	49	53	6.701	0.0096
HPV53	31	25	9.32	0.0023
HPV56	11	16	0.264	0.6074
HPV58	25	31	1.852	0.1735
HPV59	14	24	0.011	0.9156
HPV66	8	9	0.905	0.3414
HPV68	4	6	0.005	0.9461
HPV82	1	4	0.079	0.7788
HPV6	13	7	7.353	0.0067
HPV11	5	1	3.966	0.0464
HPV40	2	0	-	0.1297
HPV42	4	0	4.607	0.0318
HPV43	13	7	7.353	0.0067
HPV44	10	13	0.5605	0.4541
HPV55	5	6	0.1145	0.7351
HPV61	36	31	9.403	0.0022
HPV81	8	5	2.664	0.1026

HPV83	2	3	0.079	0.7788
-------	---	---	-------	--------

426 *P* value less than 0.05 were bold.

A

Total=790

- *U. parvum* or *U. urealyticum* 75.1%
- *M. hominis* 9.5%
- *U. parvum* or *U. urealyticum* & *M. hominis* 10.6%
- *T. vaginalis* 0.6%
- *M. genitalium* 0.1%
- *C. trachomatis* 0.1%
- HSV-2 0.1%
- *U. parvum* or *U. urealyticum* & *T. vaginalis* 1.1%
- *M. hominis* & *T. vaginalis* 0.9%
- *U. parvum* or *U. urealyticum* & *M. genitalium* 0.6%
- *U. parvum* or *U. urealyticum* & *C. trachomatis* 0.3%
- *U. parvum* or *U. urealyticum* & *M. hominis* & *M. genitalium* 0.1%
- *U. parvum* or *U. urealyticum* & *M. hominis* & HSV-2 0.1%
- *U. parvum* or *U. urealyticum* & *N. gonorrhoeae* 0.3%
- *U. parvum* or *U. urealyticum* & *M. hominis* & *T. vaginalis* 0.5%

B

Manuscript ID: 00237-25

Title: Establishment of a Two-Dimensional PCR Method for Simultaneous Detection of Nine Sexually Transmitted Disease Pathogens: Insights into Coinfection Rates and Epidemiological Trends in HPV Screening

Reviewer #1 (Comments for the Author):

Good study, but needs revision regarding the 9 different pathogens.

Reply: We sincerely appreciate the reviewer's positive feedback and constructive suggestions. In accordance with the recommendations, we have supplemented the full names of the nine pathogens upon their first mention as "9 STDs". Additionally, we have revised the two terminology errors pointed out by the reviewer. All modifications have been highlighted in yellow for clarity. Thank you for your valuable input, which has significantly enhanced the clarity and accuracy of our manuscript.

Reviewer #2 (Comments for the Author):

This study highlights the role molecular diagnostics can play in diagnosing , difficult to culture organisms.

Reply: We thank the reviewer for positive assessment and for highlighting the significance of our study. We fully agree that molecular approaches, such as the two-dimensional PCR method developed here, are crucial for the detection of fastidious or difficult-to-culture organisms, which are often missed by traditional diagnostic methods. This not only enhances diagnostic accuracy but also provides valuable insights into coinfection rates and epidemiological trends. We appreciate the reviewer's recognition of this aspect, which underscores the broader applicability of our findings in improving clinical diagnostics and public health strategies.

In addition, we have revised the manuscript to comply with the journal's formatting requirements by implementing the following changes:

restructuring the abstract into a two-part format (Abstract and Importance), standardizing reference formatting, and repositioning the “Materials and Methods” section after the “Discussion”. We believe these revisions will better align the manuscript with the journal’s academic formatting guidelines.

Re: Spectrum00237-25R1 (Establishment of a Two-Dimensional PCR Method for Simultaneous Detection of Nine Sexually Transmitted Disease Pathogens: Insights into Coinfection Rates and Epidemiological Trends in HPV Screening)

Dear Dr. Shuang Yao:

Your manuscript has been accepted, and I am forwarding it to the ASM production staff for publication. Your paper will first be checked to make sure all elements meet the technical requirements. ASM staff will contact you if anything needs to be revised before copyediting and production can begin. Otherwise, you will be notified when your proofs are ready to be viewed.

Sincerely,
Day-Yu Chao
Editor
Microbiology Spectrum

Reviewer #1 (Comments for the Author):

Good study

[revised manuscript text omitted]

progression of cervical cancer and other reproductive system diseases(4). Therefore, this study has
established an economical and simple detection method that repurposes "waste" cervical brush
samples obtained after HPV screening. This method is capable of simultaneously testing 9 types of
STDs from extracted DNA samples, including *Ureaplasma parvum* (*U. parvum*) / *Ureaplasma*
*urealyticum* (*U. urealyticum*), *Mycoplasma hominis* (*M. hominis*), *Trichomonas vaginalis* (*T.*
*vaginalis*), *Mycoplasma genitalium* (*M. genitalium*), *Neisseria gonorrhoeae* (*N. gonorrhoeae*),

[revised manuscript text omitted]

**Figure Legends**

**Figure 1. The 2D-PCR melting curves for the simultaneous detection of nine STDs in a single**
**tube.**

**Figure 2. Melting curves of different dilution plasmids for the detection of 9 STDs using the**
**2D-PCR method.**

**Figure 3. The distribution of pathogenic microorganisms among populations infected with STD**
**and their age stratification.**

**Table 1. Analysis of consistency in identification results between 2D-PCR and triple real-time**
 **PCR.**

STDPS	2D-PCR	Triple real-time PCR		Kappa
		Positive	Negative	
U. parvum / U. urealyticum	Positive	696	4	0.91
	Negative	81	1412	
M. hominis	Positive	164	5	0.98
	Negative	0	2024	
T. vaginalis	Positive	20	4	0.91
	Negative	0	2169	
M. genitalium	Positive	7	0	0.82
	Negative	3	2183	
N. gonorrhoeae	Positive	2	0	1
	Negative	0	2191	
C. trachomatis	Positive	3	0	1
	Negative	0	2190	
HSV-2	Positive	2	0	1
	Negative	0	2191	
Total	Positive	755	14	0.90
	Negative	85	1339	

**Table 2. Correlation analysis of different STDPs and HPV co-infection.**

STDPs		HPV Positive (n=516)	HPV Negative (n=1677)	χ^2	P -value
U. parvum/ U. urealyticum	Positive	206 (39.92%)	494 (29.50%)	19.89	P < 0.0001
	Negative	310 (60.08%)	1183 (70.50%)		
M. hominis	Positive	73 (14.10%)	99 (5.90%)	37.10	P < 0.0001
	Negative	443 (85.90%)	1578 (94.10%)		
T. vaginalis	Positive	7 (1.40%)	17 (1.00%)	0.43	P =0.5127
	Negative	509 (98.60%)	1660 (99.00%)		
M. genitalium	Positive	2 (0.39%)	5 (0.30%)	0.10	P =0.7528
	Negative	514 (99.61%)	1672 (99.70%)		
N. gonorrhoeae	Positive	2 (0.39%)	0 (0%)	-	P =0.0553
	Negative	514 (99.61%)	1677 (100%)		
C. trachomatis	Positive	1 (0.19%)	2 (0.12%)	-	P =0.5530
	Negative	515 (99.81%)	1675 (99.88%)		
HSV-2	Positive	1 (0.19%)	1 (0.06%)	-	P =0.4153
	Negative	515 (99.81%)	1676 (99.94%)		
STDPs	Positive	239 (46.32%)	551 (32.86%)	31.03	P < 0.0001
	Negative	277 (53.68%)	1126 (67.14%)		

*P* value less than 0.05 were bold.

**Table 3. Correlation analysis between infections of different STDPs and infections of high-risk**
 **and low-risk HPV types.**

STDPs		HR-HPV (n=353)	LR-HPV (n=110)	HLR-HPV (n=53)	χ^2	P -value
U. parvum/	Positive	124 (35.13%)	55 (50.00%)	27 (50.94%)	10.73	P=0.0047
U. urealyticum	Negative	229 (64.87%)	55 (50.00%)	26 (49.06%)		
M. hominis	Positive	45 (12.75%)	18 (16.36%)	10 (18.87%)	1.986	P =0.3704
	Negative	308 (87.25%)	92 (83.64%)	43 (81.13%)		
T. vaginalis	Positive	4 (1.13%)	3 (2.73%)	0 (0%)	2.405	P =0.3004
	Negative	349 (98.87%)	107 (97.27%)	53 (100%)		
M. genitalium	Positive	1 (0.28%)	0 (0%)	1 (1.89%)	3.613	P =0.1642
	Negative	352 (99.72%)	110 (100%)	52 (98.11%)		
N. gonorrhoeae	Positive	1 (0.28%)	1 (0.91%)	0 (0%)	1.081	P =0.5826
	Negative	352 (99.72%)	109 (99.09%)	53 (100%)		
C. trachomatis	Positive	0 (0%)	1 (0.91%)	0 (0%)	3.698	P =0.1574
	Negative	353 (100%)	109 (99.09%)	53 (100%)		
HSV-2	Positive	0 (0%)	1 (0.91%)	0 (0%)	3.698	P =0.1574
	Negative	353 (100%)	109 (99.09%)	53 (100%)		
STDPs	Positive	147 (41.64%)	61 (55.45%)	31 (58.49%)	9.954	P=0.0069
	Negative	206 (58.36%)	49 (44.55%)	22 (41.51%)		

*P* value less than 0.05 were bold.

**Table 4. Prevalence of HPV types in patients tested positive and negative for STDPs.**

HPV types	STDPs		χ^2	P-value
	Positive (n=790)	Negative (n=1403)		
HPV16	28	33	2.656	0.1031
HPV18	8	12	0.139	0.7098
HPV31	5	6	0.115	0.7351
HPV33	3	8	0.085	0.7708
HPV35	6	5	0.937	0.3330
HPV39	11	18	0.0464	0.8295
HPV45	2	4	0.0831	0.7731
HPV51	8	18	0.315	0.5745
HPV52	49	53	6.701	0.0096
HPV53	31	25	9.32	0.0023
HPV56	11	16	0.264	0.6074
HPV58	25	31	1.852	0.1735
HPV59	14	24	0.011	0.9156
HPV66	8	9	0.905	0.3414
HPV68	4	6	0.005	0.9461
HPV82	1	4	0.079	0.7788
HPV6	13	7	7.353	0.0067
HPV11	5	1	3.966	0.0464
HPV40	2	0	-	0.1297

HPV42	4	0	4.607	0.0318
HPV43	13	7	7.353	0.0067
HPV44	10	13	0.5605	0.4541
HPV55	5	6	0.1145	0.7351
HPV61	36	31	9.403	0.0022
HPV81	8	5	2.664	0.1026
HPV83	2	3	0.079	0.7788

*P* value less than 0.05 were bold.

A

Total=790

- *U. parvum* or *U. urealyticum* 75.1%
- *M. hominis* 9.5%
- *U. parvum* or *U. urealyticum* & *M. hominis* 10.6%
- *T. vaginalis* 0.6%
- *M. genitalium* 0.1%
- *C. trachomatis* 0.1%
- HSV-2 0.1%
- *U. parvum* or *U. urealyticum* & *T. vaginalis* 1.1%
- *M. hominis* & *T. vaginalis* 0.9%
- *U. parvum* or *U. urealyticum* & *M. genitalium* 0.6%
- *U. parvum* or *U. urealyticum* & *C. trachomatis* 0.3%
- *U. parvum* or *U. urealyticum* & *M. hominis* & *M. genitalium* 0.1%
- *U. parvum* or *U. urealyticum* & *M. hominis* & HSV-2 0.1%
- *U. parvum* or *U. urealyticum* & *N. gonorrhoeae* 0.3%
- *U. parvum* or *U. urealyticum* & *M. hominis* & *T. vaginalis* 0.5%

B